# The perception of shape from shading in a new light

Michael J. Proulx

Crossmodal Cognition Laboratory, Department of Psychology, University of Bath, UK

## ABSTRACT

How do humans see three-dimensional shape based on two-dimensional shading? Much research has assumed that a 'light from above' bias solves the ambiguity of shape from shading. Counter to the 'light from above' bias, studies of Bayesian priors have found that such a bias can be swayed by other light cues. Despite the persuasive power of the Bayesian models, many new studies and books cite the original 'light from above' findings. Here I present a version of the Bayesian result that can be experienced. The perception of shape-from-shading was found here to be influenced by an external light source, even when the light was obstructed and did not directly illuminate a two-dimensional stimulus. The results imply that this effect is robust and not low-level in nature. The perception of shape from shading is not necessarily based on a hard-wired internal representation of lighting direction, but rather assesses the direction of lighting in the scene adaptively. Here, for the first time, is an experiential opportunity to see what the Bayesian models have supported all along.

## INTRODUCTION

> A person entering into a room perceives, at a single glance, whence the light comes which illuminates the objects before him; and that without remaining conscious for a moment that he has attended to the circumstance: but the effect remains, and will influence his judgment.
> (*Rittenhouse, 1786*) [pp 38-39].

A number of fascinating illusions arise from graded changes in lumination. Some of the most striking give rise to the appearance of light emitting from a constant image, such as the glare effect (*Zavagno, 1999*) or the breathing light illusion (*Gori & Stubbs, 2006*). A careful arrangement of light and shadow, or a steady change in luminance, can mimic natural light in surprising ways and even represent three-dimensional shape based on shading cues alone. For example, how does one see three-dimensional shape based on two-dimensional shading, such as that shown in Fig. 1A? Initial research found that the brain assumes a single 'light from above' illuminating the discs (*Ramachandran, 1988*), suggesting that the brain is hard-wired by evolution to assume a light source that is overhead like the sun (*Sun & Perona, 1998*). Turning the figure by 90 degrees weakens the 3-D impression. More recent research found that not only is there a light from above,

Corresponding author
Michael J. Proulx,
mproulxbath@gmail.com

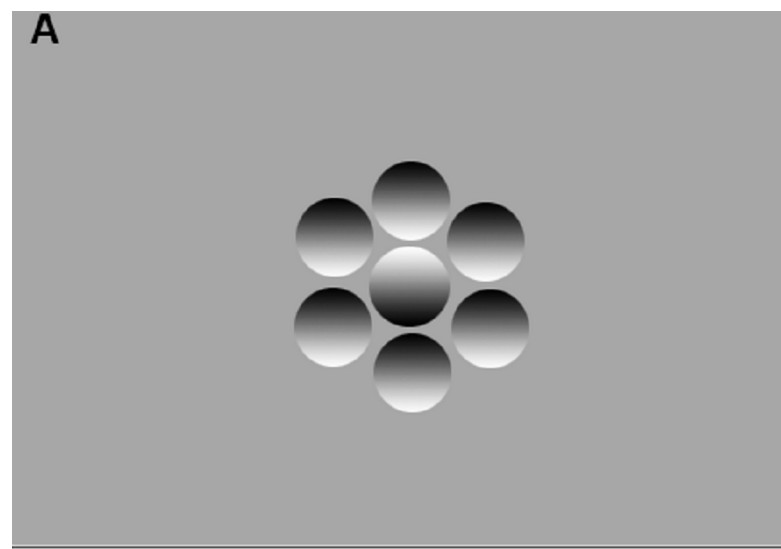

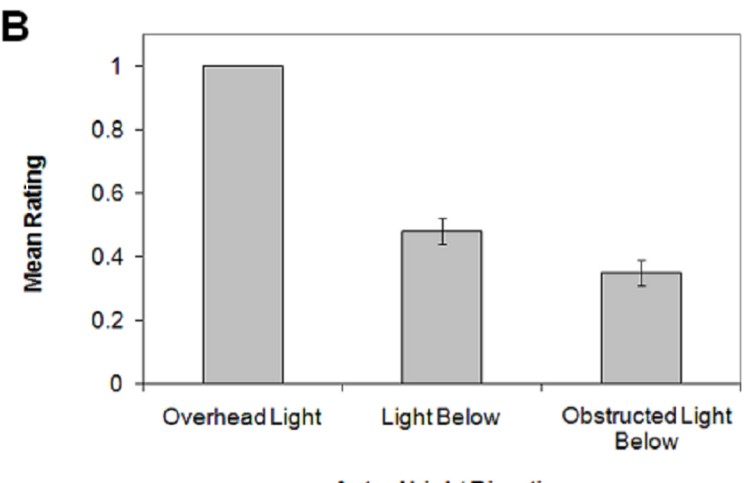

**Figure 1 Stimulus and results.** (A) An example stimulus. (B) Mean rating of perceived light direction versus the actual illumination direction in the room, as a function of experimental condition (Condition 1: Overhead Light; Condition 2: Light Below; Condition 3: Obstructed Light Below). 1.0 = light from above; 0.0 = light from below. Error bars are ±1 s.e.m.

and to the left (*Sun & Perona, 1998*). Surprisingly, research with chimpanzees revealed an assumption of a light source coming from the side; unlike humans, turning the figure by 90 degrees strengthens the 3-D percept in chimpanzees (*Tomonaga, 1998*). Have humans and chimpanzees evolved different neural assumptions about an inferred light location?

The 'single light from above' hypothesis (*Ramachandran, 1988*; *Hoffman, 1998*) is the most favoured perspective amongst researchers in recent publications (*Thomas, Nardini & Mareschal, 2010*; *Gerardin, Kourtzi & Mamassian, 2010*; *Dobbins & Grossmann, 2010*). Relying instead on an external light source (*Yonas, Kuskowski & Sternfels, 1979*) would make adaptive sense: rather than having an assumption that might come into conflict with an actual source of object shading, one should take advantage of an external light cue

when interpreting scenes. Any conflict between the actual light source and that perceived in computer graphics could slow responses and reduce accuracy.

Although most recent investigators have accepted the "light is overhead" bias as a strong rule used by the human visual system (*Thomas, Nardini & Mareschal, 2010*; *Gerardin, Kourtzi & Mamassian, 2010*; *Dobbins & Grossmann, 2010*), a number of studies have taken a Bayesian modelling approach to demonstrate how other information is incorporated with such a prior to modify the ultimate perception in favour of the most likely outcome (*Stone, Kerrigan & Porrill, 2009*; *Stone, 2011*; *Adams, 2007*; *Adams, 2008*; *Adams, Graf & Ernst, 2004*). Contrary to those who theorized that the overhead bias is hard-wired and innate (*Ramachandran, 1988*; *Hoffman, 1998*), Stone found that the overhead Bayesian prior requires developmental experience (*Stone, 2011*). A recent study demonstrated that the assumption that light comes from above has a lesser role than lighting cues in the perception of shape from shading (*Morgenstern, Murray & Harris, 2011*). Morgenstern and colleagues (*2011*) presented lighting cues that could influence shape perception in either accordance with an objects context in the image or instead with the light from above prior. The light direction implied by the context had a greater effect on shape from shading perception. Also, little notice has been given to aspects of previous research (*Yonas, Kuskowski & Sternfels, 1979*; *Berbaum, Bever & Chung, 1983*) that suggest the overhead bias can be overridden by an external light source. *Yonas, Kuskowski & Sternfels (1979)* have most often been cited for demonstrating that children primarily perceived shape from shading in egocentric "light is overhead" coordinates. However, they also reported that a significant proportion of the subjects perceived the stimuli in reference to actual external light as well (though at a lower proportion than those using the egocentric reference frame). What is remarkable about these results is that even though the light source was covered (thus perhaps making its location ambiguous), a significant number of the children perceived the stimuli as shaded consistent with the light's location. This also points to possible individual differences in a seemingly simple, low-level task, has others have noted as well (*Koenderink, van Doorn & Kappers, 1992*; *Todd et al., 1996*).

Berbaum and colleagues (*1983*) examined the impact of the direction of illumination on the visual interpretation of an actual, three-dimensional muffin pan. Subjects viewed the pan with the light coming from below; by incorporating mirrors, however, the experimenters had it appear to the subjects that the light was actually coming from above. The subjects perceived the shape based on the remembered light source direction (the light source could not actually be seen at the same time as the stimulus). The extension of this work to two-dimensional stimuli has been questioned (*Kleffner & Ramachandran, 1992*) because it had not been replicated in previous work (*Ramachandran, 1988*).

However, there are two inaccuracies to this criticism by *Kleffner & Ramachandran (1992)*. First, the study by *Berbaum, Bever & Chung (1983)* was incorrectly cited as having had the subjects discover the actual light source direction by putting out their hands to cast a shadow, with a resulting inversion of relief when the light source direction was discovered. In fact, the *Berbaum, Bever & Chung (1983)* study put sticks on the muffin pans to cast shadows that revealed the true direction of illumination and they found that this

did *not* cause a reversal of relief (the perception continued to be based on the direction of remembered illumination direction). Second, the previous study by *Ramachandran (1988)* has its own drawbacks as well in terms of addressing the impact of illumination direction. The direction of illumination was itself ambiguous: a concave mask of a face, although lit from above, is perceived as a normal face lit from below; thus the direction of illumination was assumed by the experimenter to be from below (consistent with the face perception) however subjects still viewed the shape from shading discs as if they were lit from above (consistent with the actual direction of illumination on the convex mask that appeared as a concave face).

Here we tested a demonstration that the visual system might rely more on an external light cue to judge shape from shading with a straightforward demonstration in the spirit of *Ramachandran (1988)*. Given the evidence presented by a number of studies, summarized and tested by Morgenstern and colleagues (*2011*), it should be well known now that the original finding of a light from overhead bias is not the whole story. Despite such evidence to the contrary, the light from overhead view is still often cited (*Thomas, Nardini & Mareschal, 2010*; *Gerardin, Kourtzi & Mamassian, 2010*; *Dobbins & Grossmann, 2010*). Here we found a robust version of the shape from shading experiment that can be experienced even with a printed photograph of a shape from shading stimulus and an external light cue, thus rendering the effect in manner suitable for a sceptical reader or a classroom setting.

## METHODS

### Subjects

Naive observers viewed stimuli from a distance of 50 cm, and under only one of the three conditions: overhead lighting ($n = 5$); a light below ($n = 14$); or an obstructed light below ($n = 9$). All observers had normal or corrected-to-normal visual acuity (wearing glasses or contact lenses if necessary). Ethical approval was obtained from the JHU IRB, QMREC and Bath Psychology Ethics Committee, and all participants gave informed, written consent.

### Stimuli & procedure

The reader can replicate the effect used here in a dark room with a single light placed below the image in Fig. 1A. The discs (2 cm in diameter) had a vertical luminance gradient from white at one end to black at the other, with the centre of the disc at the same level of grey as the background, and were printed on standard (flat) white paper.

In conditions with the light from below, the room was completely dark except for an incandescent light (30 W) placed below the stimuli, except for the condition where a thin board obstructed the entire beam of light. The direction of the actual light source was a between-subjects manipulation to avoid drawing attention to the light. The observers told the experimenter whether the different discs appeared either as bumps or cavities, and were never asked about any real or implied light source. Note that there were a total of 36 discs shown on seven different trials each with a different configuration of 1–12 discs to each observer. The observers were asked to point to each disc present for each trials and report if it appeared as a bump or a cavity. All observers reported the discs as bumps or cavities

in a manner consistent with perceiving only one light source; that is, no participants ever reported discs with opposite polarity of shading as both being bumps or both being cavities. Each configuration of discs was shown to each participant once.

The stimuli were printed on paper so all competing light cues (including a back-lit monitor) were removed and the shape from shading of the discs was completely ambiguous, unlike previous work (*Berbaum, Bever & Chung, 1983*). The responses for each disc were converted to light direction ratings to reflect whether observers perceived a stimulus as being lit from above or from below. The light direction was inferred from their judgments of disc convexity. For example, if a disc was labelled a "bump" and its shading was white at the top, and black at the bottom, then this indicated that the perceived direction of illumination was from above because a bump would be brightest at the portion nearest the illumination. If the perceived direction of illumination was above, then the response was recorded as "1"; if the perceived direction of illumination was below, then the response was recorded as "0". Thus, if all observers perceived every disc as being lit from overhead, then the average rating of illumination direction would be 1. Thus the analysis is similar to that of assessing accuracy as a proportion, however here computed as a function of how each disc was classified. All analyses were based on these ratings, averaged across observers and trials. The null hypothesis on the basis of the literature and the overhead light condition is that all ratings should be rated as 1.0, consistent with the 'light from above' assumption (*Ramachandran, 1988*). To provide a directional test of the hypothesis a one-tailed $t$-test was used to evaluate the results; however exact $p$ values are reported also.

## RESULTS

### Overhead light

In the overhead lighting condition, all participants reported all stimuli as bumps or cavities in a manner consistent with the 'light from above' assumption, resulting in a rating of 1.0 (see Fig. 1B, 'Overhead Light'), replicating the original finding (*Ramachandran, 1988*).

### Light below

Figure 1B shows that, when the external light source was below the images, the average proportion was only 0.48 (see 'Light Below'); this indicates that the stimuli were often perceived in a manner consistent with the external light and inconsistent with the 'light from above' assumption, $t(13) = -6.18$, $P = .00003$, $\eta^2 = 0.718$ (one-tailed $t$-test). Individual differences were primarily responsible, with six out of 14 observers' responses consistent with the light source.

Two possible mechanisms might be responsible for the effect of the external light: first, a low-level process (*Wenderoth & Hickey, 1993*) whereby cues created by the external light interact with the grain of the paper on which the stimuli were presented; or second, a higher level process, whereby a representation of the physical location of the light source is responsible for the influence of an external light (*Yonas, Kuskowski & Sternfels, 1979*; *Berbaum, Bever & Chung, 1983*).

### Obstructed light below

To distinguish between these mechanisms, a third condition featured a light source again placed below the stimuli, however a board was placed between the light and the stimuli, and thus the light was indirect. If the previous results were due to low-level shading cues, then the average rating should be closer to 1.0, consistent with the 'light from above' assumption. In fact, when the light was obstructed such that there were no low-level light cues on the stimulus (see 'Obstructed Light Below'), the average rating was only 0.35; this means that stimuli were again perceived in a manner consistent with the external light cue, $t(8) = -6.61, P = .0002, \eta^2 = 0.845$ (one-tailed $t$-test). Seven out of nine of the observers were primarily affected by the light source. There was not a significant difference between the unobstructed and obstructed light below conditions.

## DISCUSSION

The results are inconsistent with the hypothesis that the visual system relies exclusively on the 'light from above' assumption, and furthermore demonstrate that this effect is not low-level in nature, consistent with past work on this topic that found a memorial representation of light location was used to interpret the shape of an actual muffin tin (*Berbaum, Bever & Chung, 1983*) even when the light was occluded and did not directly illuminate a two-dimensional stimulus. Note that similar to previous research by Yonas and colleagues (*1979*) with children, individual differences in the perception of shape-from-shading are present, and future research will better reveal the source of this diversity. The results further imply that this effect is not low-level in nature, and thus consistent with the quote from Rittenhouse's initial article the described shape from shading for the first time (*Rittenhouse, 1786*).

The present findings provide converging support with recent behavioural evidence that lighting cues, rather than a 'light from above' bias, are the primary determinant of the perception of shape from shading (*Morgenstern, Murray & Harris, 2011*). The results reported here provide further evidence that the perception of shape from shading, like other aspects of visual perception (*Purves, Wojtach & Lotto, 2011*), arises from interactions with the natural world rather than internal biases that give rise to illusory perception (*Stone, 2011*; *Stone & Pascalis, 2010*). Importantly the demonstration here does not depend on a complex analysis of psychophysical data, but rather a simple shape-from-shading drawing and a candle would do, allowing even *Rittenhouse (1786)* to experience this himself.

## ACKNOWLEDGEMENTS

I thank Christopher Min and Monique Green for assistance, and Howard Egeth for helpful discussions.

### Funding

This study was supported in part by the National Science Foundation (GRF). The funders had no role in study design, data collection and analysis, decision to publish, or preparation of the manuscript.

### Grant Disclosures

The following grant information was disclosed by the author:
National Science Foundation (GRF).

### Competing Interests

The author declares there are no competing interests.

### Author Contributions

- Michael J. Proulx conceived and designed the experiments, performed the experiments, analyzed the data, contributed reagents/materials/analysis tools, wrote the paper, prepared figures and/or tables, reviewed drafts of the paper.

### Human Ethics

The following information was supplied relating to ethical approvals (i.e., approving body and any reference numbers):

Ethical approval was obtained from the JHU IRB, QMREC and Bath Psychology Ethics Committee, and all participants gave informed, written consent.

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
