# Peer review of "The perception of shape from shading in a new light"

_PeerJ, doi:10.7717/peerj.363_

## Round 0.1 · original submission · Minor Revisions

· Academic Editor

Minor Revisions

The reviewers both liked the study and provided clear and important suggestions for revision. Please do address these fully as I feel they are important to the final successful publication of the study.

·

Basic reporting

See below

Experimental design

See below

Validity of the findings

See below

Additional comments

How many times was each stimulus presented to each subject?

The mean score for one group was 1 (ie at ceiling) and for the other was 0.48. I doubt a t-test is the correct test to use here, esp with such low subject numbers and (single?) binary response from each subjects. 6/14 subjects saw figure consistent with light source direction.

The method is unclear. Which disc were subjects asked to respond to in the stmulus, and how many times was the stimulus presented?

Interesting results. I did a similar experiment a few years ago and found no effect of the postiion (above/below) of a light on perceived shape at all (thus, unpublished).

It would be good to have more details of the method and data collected, and if the statistical tests used are appropriate.

Some relevant references

Stone J (2011) Footprints sticking out of the sand (Part II): Children’s Bayesian priors for lighting direction and convexity. Perception 40: 175-190.

Stone, J., Kerrigan, I., and Porrill, J. (2009). Where is the light? Bayesian perceptual priors for lighting direction. Proceedings Royal Society London (B), 276:1797–1804.

Reviewer 2 ·

Basic reporting

no comment

Experimental design

no comment

Validity of the findings

no comment

Additional comments

The submitted manuscript entitled “The perception of shape from shading in a new light” is a very well written research paper composed by an experiment with clear results. However, few more efforts should be done to reach the final version ready to be publish. Here are my suggestions:

1) I would suggest to do a more general introduction about the illusory effect produced by stimuli that resembled light source and shadow like the glare effect (Zavagno, 1999) the breathing light illusion (Gori and Stubbs, 2006) and the Checker Shadow Illusion (Adelson, 1995) it will put the paper in a more general context that will attract readers.
2) When the Author wrote: “A number of studies have taken a Bayesian modelling approach” few examples of those studies should be cited.
3) After “Morgenstern and colleagues” the number of the citation should be put between square brackets.
4) After “Yonas et al” the number of the citation should be put between square brackets.
5) After “Kleffner and Ramachandran” the number of the citation should be put between square brackets.
6) After “Berbaum et” (two times, line 80 and 82) the number of the citation should be put between square brackets.
7) After “Ramachandran” the number of the citation should be put between square brackets.
8) In this sentence “light from overhead bias not the whole story” an “is” is probably missed.
9) In the method it should be specified if the observers had normal or corrected-to-normal visual acuity.
10) In the method the power of the incandescent light should be reported, I’m aware that it could not be crucial for the effect but it could still be nice to report that for future replications.
11) When the Author wrote “Yonas and colleagues (1979)” the year should be changed with the number of the citation between square brackets.
12) When the Author wrote “Rittenhouse’s initial article described shape from shading for the first time (1786) ” the year should be changed with the number of the citation between square brackets.

---

## Round 0.2 · accepted · Accept

· Academic Editor

Accept

Thank you for your submission. It's a nice study!